# Reproducibility of Explaining Groups of Points in Low-Dimensional Representations

**Ruud van Bakel**
Master Artificial Intelligence
University of Amsterdam
ruud.van.bakel@student.uva.nl

**Abhijith Chintam**
Master Artificial Intelligence
University of Amsterdam
abhijith.chintam@student.uva.nl

**Andreas Hadjipieris**
Master Artificial Intelligence
University of Amsterdam
andreas.hadjipieris@student.uva.nl

**Roel Kuiper**
Master Artificial Intelligence
University of Amsterdam
roel.kuiper@student.uva.nl

## Reproducibility Summary

**Scope of Reproducibility**

The main claims of the paper Explaining Groups of Points in Low-Dimensional Representations (Plumb et al., 2020) include introduction of a new type of explanation - Global Counterfactual Explanation (GCE) which is relatively sparse and is consistent i.e., symmetrical and transitive among all the groups. The paper also claims that the explanations based on Transitive Global Translations (TGT) algorithm are better than Difference Between the Mean (DBM) baseline for varying degrees of sparsity. The TGT algorithm is also claimed to be capturing the real signals in the data.

**Methodology**

For reproducing the paper we decided to convert the Tensorflow 1.0 implementation of the authors to a PyTorch implementation. We also reproduced the results from the original code base to verify the primary claims of the paper. We also replicated the results for a new Crop mapping using fused optical-radar dataset (Dua and Graff, 2017).

**Results**

We were successfully able to reproduce the results of the paper. The results of TGT's coverage, correctness and sparsity in comparison with DBM were similar to those claimed by the authors on all datasets.

**What was easy**

It was reasonably easy to run the authors' original Tensorflow code. The authors have have clearly separated the experiments on different datasets from the core implementation of the algorithm which helped greatly while re-implementing in Pytorch. Additionally, the notebooks provided for each of the experiments streamlined reproducing the results.

**What was difficult**

A Readme file explaining the steps to setting up and running the code would have made it a much better experience. We faced two challenges in particular while running the authors' code. Since the authors had some fixed paths to files in the code base, running the code required minor adjustments. Additionally, the python library dependencies were not clearly mentioned, these could have been provided as environment details. For example the scvis (Ding, 2018) library was used but not mentioned in the readme of the code base.

**Communication with original authors**

Communication with the original author's was not necessary to reproduce the code.

33rd Conference on Neural Information Processing Systems (NeurIPS 2020), Vancouver, Canada.

# 1    Introduction

The central focus of the original paper Explaining Groups of Points in Low-Dimensional Representations (Plumb et al., 2020) is to introduce a novel way to explain groups of points in low-dimensional representations by leveraging the model that produced them in the first place. The authors relied on counterfactual explanations (Verma et al., 2020), a specific class of explanation that provides a link between what could have happened had input to a model been changed in a particular way.

The goal of the authors was to find an approach that can generate interpretable and effective explanations which are global and consistent. Intuitively, a consistent explanation is both symmetric and transitive. The authors used coverage and correctness metrics to measure the effectiveness of the explanation. For an explanation to be interpretable, it should be sparse and yet be effective. The paper proposes a novel and interesting approach to solve the interpretable machine learning problem. In this work, we tried to reproduce the results and verify the claims of the authors and also experiment on alternative datasets and provide a pytorch implementation of the algorithms proposed by the authors.

# 2    Scope of reproducibility

This paper introduces a new type of explanation - Global Counterfactual Explanation (GCE) which is relatively sparse and is consistent i.e., symmetrical and transitive among all the groups. The authors with their experiments on different datasets show that their Transitive Global Translations (TGT) algorithm is better than Difference Between the Mean (DBM) baseline for varying degrees of sparsity in terms of both coverage and correctness which are the two metrics used by the authors to measure the quality of the explanations. The TGT algorithm is also claimed to have captured the real signals in the data.

At the outset, we decided to reproduce and verify the following claims of the paper.

- Global Counterfactual Explanations are effective, global and consistent on different datasets.
- GCEs are more interpretable as they can provide sparse explanations than the baseline methods.
- Transitive Global Translations algorithm captures the real signals in the data.

# 3    Methodology

For reproducing the paper, we followed a 4-step incremental approach. Firstly, we decided to run the code base in Tensorflow 1.x published by the authors and reproduce the results claimed in the paper. Our second step was to see if we can reproduce the general claims made by the authors by changing parameters and settings used by the the authors while running their experiments. This step is important to verify if there are any selective or under reporting of results. As part of this, we experimented with various groups of points in latent space and also different sparsity levels to verify the transitivity and symmetry properties of the explanations provided by the TGT algorithm.

We further decided to replicate the claims made in the paper using a new dataset altogether. While the authors have experimented with 5 different datasets, only the Single-Cell RNA dataset is very rich in terms of number of features and instances. The rest of the datasets as described in Section 3.2 have less number of features and instances. We thought it is important to test the TGT algorithm on an alternate dataset with sufficiently large feature set to validate the claims on the sparsity and similarity of the explanations meaningfully. We also preferred a multi-class dataset for this task as it would be ideal for testing the transitive explanations.

Finally, we also decided to convert the Tensorflow 1.x implementation of the authors to a PyTorch implementation. We also reproduced the results from the original code base to verify the primary claims of the paper. An additional resource that is required to run the code is the scvis library (Ding, 2018).

## 3.1    Model descriptions

For reproducing the results, the authors' models are used in Tensorflow. Additionally, these models have been retrained in Pytorch to further test the reproducibility. The models used in the paper are Variational AutoEncoders (VAE), first introduced by Kingma and Welling (2014), which are adjusted to fit the different datasets. A VAE is a type of autoencoder that learns to reproduce the input while also mapping the data to a latent space. The VAE tries to reproduce its input by sampling from the latent space. Where VAE differs from normal autoencoders is in the variational part of the term, which refers to variational inference. This allows the VAE to learn distributions of the parameters instead of having to learn point estimates.

## 3.2 Datasets

For reproducing the results of the paper, we have used the datasets provided by the authors which include the UCI Iris, Boston housing and Heart disease datasets and a Single-cell RNA dataset (Shekhar et al., 2016). These datasets were included in the code base of the original paper. For proving some claims, the authors also created a synthetic dataset.

- **UCI Iris** This dataset contains information about iris plants. It has three iris classes and 50 instances for each class. Each instance consists of four features.

- **UCI Boston Housing** In this dataset are 506 instances of houses. Each instance has 13 features. The goal of this dataset is to predict the prices of the house instances. This dataset can not be found at the UCI site anymore and should be acquired from another source.

- **UCI Heart Disease** The Heart Disease dataset contains 303 instances of patients, potentially with a heart disease. Each instance consists of 14 features and the task is to predict whether or not someone has a heart disease.

- **Synthetic** This dataset contains 400 instances with 4 features. Every feature is generated by a random value that its scaled by a factor of 0.2, 0.2, 0.5, 0.05 respectively. The forth feature is a noisy version of the first feature.

- **Single-Cell RNA** This dataset contains 44,994 instances with 24,904 features. After applying PCA and a selection, 27,499 instances remain, each with 13,166 features. The observations have to be mapped to one of 18 different classes.

Since the Single-Cell RNA dataset has very high number of features and many clusters (18) as compared to other datasets and it is very appropriate for comparing the sparsity of the explanations.

We also experimented with a crop dataset to verify if the claims of the authors can be generalised to any other datasets apart from those the authors have experimented on.

- **UCI Crop Mapping** This dataset contains 32,5831 instances, each with 175 features. One of these features is the crop label, which effectively leaves 174 features. The task for this dataset is to assign each instance to one of seven crop classes. For our experiments we took a subset of this dataset by sampling 1,000 instances for each class. This also balanced the dataset, as the original had varying amounts of instances for each class (e.g. 75,673 canola instances and 1,143 broadleaf instances).

## 3.3 Hyperparameters

Sparsity and L1 Regularization

One of the desirable properties of explanations of low-dimensional representations is their interpretability. For the explanations to be interpretable they have to be sufficiently sparse. But sparsity comes at a cost because of the accuracy-interpretability trade-off where in explanations that are too sparse can not approximate a complex model well enough. The Transitive Global Translations algorithm uses a loss function with L1 regularization and the sparsity is roughly controlled by regularization hyper-parameter $\lambda$.

In the experiments, the hyper-parameter $\lambda$ is tuned separately for each sparsity level K and authors proposed a similarity metric $similarity(e_1, e_2)$ to capture how much of $e_1$'s explanation uses features that were also chosen by $e_2$ where $e_1$ and $e_2$ are explanations for different sparsity levels.

## 3.4 Experimental setup

The structure of the authors' code, given some minor tweaks, allowed for easy reproduction. Each dataset has its own notebook to run the experiment, since the models were also uploaded to their GitHub (Plumb, 2020). After converting the Tensorflow code and notebooks to Pytorch, the same procedure was applied. The code can be found at `https://github.com/antreashp/FACT_2020`.

## 3.5 Computational requirements

We run the algorithm provided by the authors on a Intel Core i7-6700HQ CPU @ 2.60 GHz and it took around 2 hours to reach completion. Since the model was shallow there was no real need for GPU support. Each experiment was using 10 GB of RAM on average.

# 4 Results

## 4.1 Result 1

The first primary claim of the paper is that GCEs are effective, global and consistent on different datasets. For explanations to be consistent, they have to be symmetric and transitive. Being symmetric means that $\delta_{i \to j}$, the explanation given by the algorithm for a point to move from group $i$ to group $j$ and $\delta_{j \to i}$, the explanation given by the algorithm for a point to move from group $j$ to group $i$ should be the same vector with opposite direction. Being transitive means that $\delta_{i \to k}$ should be close to $\delta_{i \to j} + \delta_{j \to k}$ where $i$, $j$ and $k$ are three different groups. In order to check if the explanations indeed had these properties, we tested some samples for all datasets. Testing for all explanation combinations is not feasible, since higher dimensional datasets have a rather large amount of combinations. For example, with the Single Cell RNA dataset 18! explanation combinations are possible. Almost all samples turned out to be transitive and symmetric. These properties should stand true for different sparsity levels to generalize the authors' claims. We have run the experiments on different datasets for different sparsity levels and found some cases where the explanations were not transitive as claimed by the authors.

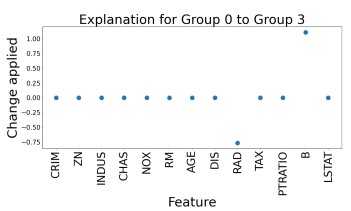

Figure 1: Housing dataset (K=2): Explanation for group 0 to group 3

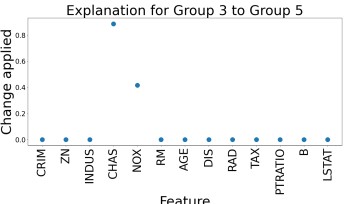

Figure 2: Housing dataset (K=2): Explanation for group 3 to group 5

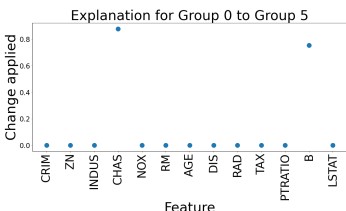

Figure 3: Housing dataset (K=2): Explanation for group 0 to group 5

But in all our experiments on different datasets, for the explanations between different pairs of groups remained symmetric as claimed by the authors.

The effectiveness of the explanations produced by TGT algorithm as measured by correctness and coverage metrics is in line with the claims made in the paper. For sparse explanations on different datasets, TGT was clearly a winner on both the metrics. But as the number of features used for the explanations increased, the coverage and correctness of both TGT and DBM converged as shown in Figure 5 in case of Bipolar dataset.

## 4.2 Result 2

We experimented to verify second claim of the authors in the paper that GCEs are more interpretable as they can provide sparser explanations than the DBM. We reproduced the explanations for different datasets at varying levels of sparsity by tuning $\lambda$ appropriately. The explanations are compared for coverage, correctness and similarity metrics at different levels of sparsity. On a whole, TGT performed well compared to DBM in terms of coverage and correctness metrics. While DBM by definition has a similarity of 1 for any pair of sparsity levels, we could reproduce that TGT is fairly consistent at picking a subset of the current features when asked to find an even sparser solution.

On all the datasets, we could reproduce that TGT outperforms DBM both on coverage and correctness for lower number of features. As claimed by the authors, for the single-cell RNA dataset, TGT significantly as shown in the Figure 5 outperforms for less than 250 features. But as the number of features increase, DBM seems to outperform TGT albeit by a small margin. We reproduced similar pairwise explanation metrics for different groups and TGT clearly produced effective explanations with high coverage and correctness for different pairs of groups.

As laid down in our approach to reproducibility of the paper, we delved a step further and conducted all the experiments done by the authors on our new dataset UCI Crop Mapping (Dua and Graff, 2017). For the experimental purposes, we have pre-processed the data by normalizing and balancing by random sampling. The results were quite similar to those on other datasets. The TGT algorithm could clearly produce more effective explanations than DBM baseline for lower sparsity levels as shown in Figure 4. But the similarity metric deviated from the claims of the authors and dropped below 0.9 unlike the datasets used by the authors where it was fairly close to the DBM baseline.

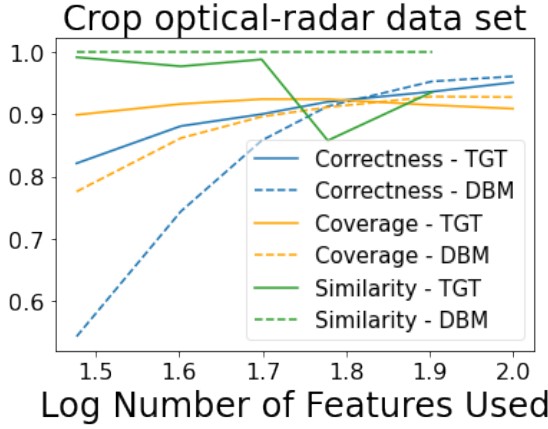

Figure 4: TGT vs DBM for Crop dataset

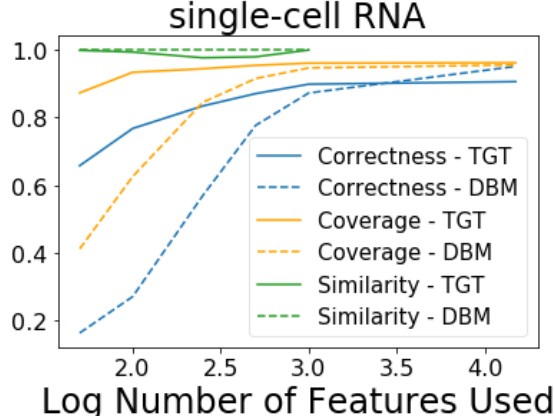

Figure 5: TGT vs DBM for Bipolar dataset

### 4.3 Result 3

The last primary claim of the paper is that the TGT algorithm captures the real signals in the data. Taking into account that result 2 shows that TGT provides accurate explanations, in practice there could be a mismatch between the structure of the data and what the model learns. The authors have defined an experiment in which the TGT identifies the structure of a synthetic dataset, of which the causal structure is known. We were able to reproduce the explanations from the TGT for the synthetic dataset. These explanations showed that the explanations provided by the TGT successfully ignored the noise added to the dataset while DBM was not able to find an explanation that was independent of the added noise.

Secondly, the authors compared TGT's explanations against basic domain knowledge on the UCI datasets. An analysis of the UCI Datasets using the labels demonstrates that TGT is able to find the explanations that match with the domain knowledge. For example, for the Iris dataset groups are separated based on the Petal Width feature. Additionally, we could reproduce for the remaining UCI datasets TGT's explanations are also consistent with the domain knowledge.

Last of all, the authors adjusted and translated the UCI datasets to generate more data to be able to do a quantative analysis. We were successfully able to reproduce that TGT correctly identifies that modifications were made to the original datasets. Additionally, the explanations do not significantly change meaning that TGT is able to find real patterns in the data.

## 5 Discussion

### 5.1 What was easy

It was reasonably easy to run the authors' original Tensorflow code. The authors have have clearly separated the experiments on different datasets from the core implementation of the algorithm which helped greatly while re-implementing in Pytorch. Additionally, the notebooks provided for each of the experiments streamlined reproducing the results.

The authors also provided trained model files and explanations making it convenient to analyse the results without retraining the model and algorithms. They provided a switch to optionally run the experiments from the scratch by training the models and evaluating the explanations.

### 5.2 What was difficult

A Readme file explaining the steps to setting up and running the code would have made it a much better experience. We faced particularly two challenges while running the authors' code. Since the authors had some fixed paths in the code base, running the code required minor adjustments. Additionally, the python library dependencies were not clearly mentioned, these could have been provided as environment details. For example the scvis library was used but not mentioned in the readme of the code base. Similarly the authors also used Integrated Gradients but it's not clearly mentioned in the readme file.

While running the experiments for the new dataset, we had to manually select the vertices for the clusters in latent space. It is not documented anywhere and had to be figured out from the code. We tried to automate this step and used Gaussian mixture model for clustering and it produced similar results.

## 5.3 Communication with original authors

Communication with the original author's was not necessary to reproduce the code.

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
