# OpenReview forum: "Explaining Groups of Points in Low-Dimensional Representations"
_ML_Reproducibility_Challenge/2020 — Reject_

### Official Review · AnonReviewer3 · 2021-02-26
**Clear reproducibility review on Explaining Groups of Points in Low-Dimensional Representations**

**Rating:** 9
**Confidence:** 4

**Review:**

This reproducibility review is of high quality: the authors were successfully reproducing the earlier results in another modeling ecosystem and the reporting is clear.

If there would be anything to improve, it would be about finding some examples that pose potential problems for reproducibility; even with good reproducibility there can be potential pitfalls and reporting what has been done to identify and point out the weak points would be informative if such examples can be found.


**Familiar With The Original Paper:**

I have not read the original paper

**Reproducibility Summary:**

Report has summary

---

### Official Review · AnonReviewer2 · 2021-02-28
**GCE + TGT**

**Rating:** 8
**Confidence:** 3

**Review:**

The authors have done a fine job in reproducing the original paper.
* The reproducibility summary is provided.
* It seemed fairly easy to reproduce the results from the paper.
 ** The authors converted the original code to Pytorch and started their analysis from scratch. Their results tallied with that of  the original paper.
 ** Further, the authors ran the code on a new Crop dataset checking code/idea generalizability. This is a great move.

* Even though no communication with the original authors was needed, some areas to tidy up in the code and notebooks are provided by the authors but these are minor.
* Overall approach to reproduce the results and the paper are clear.

**Familiar With The Original Paper:**

I have not read the original paper

**Reproducibility Summary:**

Report has summary

---

### Decision · Program_Chairs · 2021-03-31

**Decision:**

Reject

**Comment:**

Overall reviews and/or the paper content not good enough for the AC to recommend to the journal.

While this report got high scores, it was not anonymous, and could have presented a more thorough analysis of the original work.